# Hmong microbiome ANd Gout, Obesity, Vitamin C (HMANGO-C): A phase II clinical study protocol

Ya-Feng Wen[1], Kathleen A. Culhane-Pera[2,3], Shannon L. Pergament[3], Yeng Moua[3], Bai Vue[3], Toua Yang[3], Muaj Lo[2], Boguang Sun[1], Dan Knights[4,5,6], Robert J. Straka[1]*

1 Department of Experimental and Clinical Pharmacology, College of Pharmacy, University of Minnesota, Minneapolis, Minnesota, United States of America, 2 Minnesota Community Care, St. Paul, Minnesota, United States of America, 3 SoLaHmo Partnership for Health and Wellness, Community-University Health Care Center, Minneapolis, Minnesota, United States of America, 4 Bioinformatics and Computational Biology Program, University of Minnesota, Minneapolis, Minnesota, United States of America, 5 Biotechnology Institute, University of Minnesota, Minneapolis, Minnesota, United States of America, 6 Department of Computer Science and Engineering, University of Minnesota, Minneapolis, Minnesota, United States of America

* strak001@umn.edu

**Data Availability Statement:** De-identified relevant clinical data will be made available upon study completion. The minimal data set, including shotgun metagenomic sequencing data will be

## Abstract

### Background

Hmong men in Minnesota exhibit a high prevalence of gout and hyperuricemia. Although evidence of vitamin C's effectiveness as a treatment for gout is mixed, analysis of therapeutic benefit based on an individual's multiomic signature may identify predictive markers of treatment success.

### Objectives

The primary objective of the Hmong Microbiome ANd Gout, Obesity, Vitamin C (HMANGO-C) study was to assess the effectiveness of vitamin C on serum urate in Hmong adults with and without gout/hyperuricemia. The secondary objectives were to assess if 1) vitamin C impacts the taxonomic and functional patterns of microbiota; 2) taxonomic and functional patterns of microbiota impact vitamin C's urate-lowering effects; 3) genetic variations impact vitamin C's urate-lowering effects; 4) differential microbial biomarkers exist for patients with or without gout; and 5) there is an association between obesity, gut microbiota and gout/hyperuricemia.

### Methods

This prospective open-labelled clinical trial was guided by community-based participatory research principles and conducted under research safety restrictions for SARS-CoV-2. We aimed to enroll a convenient sample of 180 Hmong adults (120 with gout/hyperuricemia and 60 without gout/hyperuricemia) who provided medical, demographic, dietary and anthropometric information. Participants took vitamin C 500mg twice daily for 8 weeks and provided pre-and post- samples of blood and urine for urate measurements as well as stool samples

deposited in GenBank®, the National Institute of Health genetic sequence database.

**Funding:** This study was also supported by the University of Minnesota Clinical and Translational Science Institute, the Graduate School at the University of Minnesota, University of Minnesota and National Institutes of Health's National Center for Advancing Translational Sciences, grant UL1TR002494. The content is solely the responsibility of the authors and does not represent the official views of the National Institutes of Health's National Center for Advancing Translational Sciences.

**Competing interests:** D.K. serves as CEO and holds equity in CoreBiome, a company involved in the commercialization of microbiome analysis. The University of Minnesota also has financial interests in CoreBiome under the terms of a license agreement with CoreBiome. These interests have been reviewed and managed by the University of Minnesota in accordance with its Conflict-of-Interest policies. This does not alter our adherence to PLOS ONE policies on sharing data and materials.

for gut microbiome. Salivary DNA was also collected for genetic markers relevant to uric acid disposition.

## Expected results

We expected to quantify the impact of vitamin C on serum urate in Hmong adults with and without gout/hyperuricemia. The outcome will enhance our understanding of how gut microbiome and genomic variants impact the urate-lowering of vitamin C and associations between obesity, gut microbiota and gout/hyperuricemia. Ultimately, findings may improve our understanding of the causes and potential interventions that could be used to address health disparities in the prevalence and management of gout in this underserved population.

## Trial registration

ClinicalTrials.gov NCT04938024 (first posted: 06/24/2021).

## Introduction

Gout, often caused by chronic elevation of serum urate (SU) as hyperuricemia (HU), is the most common form of inflammatory arthritis, affecting 0.1% to 10% of the worldwide population [1]. About 3.9% of adults in the United States (U.S.) suffer from gout and prevalence is even higher in certain ethnicities [2]. Patients with gout have higher rates of cardiometabolic diseases, such as obesity, hypertension, renal disease, type 2 diabetes (T2DM), which significantly impact morbidity, mortality, and healthcare costs [3–5]. The Hmong community in Minnesota is a high-risk population, exhibiting a 2-fold higher prevalence of gout [6], 4-fold increase in tophaceous gouty arthritis [7], and a 5-fold higher prevalence of uric acid renal stones [8], as compared to non-Hmong in Minnesota. Finding a potential low-cost, safe, readily available, and effective agent to reduce serum urate in a high-risk population, even when they have not manifested with gout or hyperuricemia, could be of interest to the community as well as the clinicians who are taking care of this population.

To prevent acute gout attacks and reduce complications associated with gout and/or HU, it is critical to maintain UA below 6 mg/dL [9–11]. However, despite the availability of numerous urate-lowering therapies (ULT), up to 70% of patients fail to achieve target SU [12], and even higher treatment failure rates in select populations including the Hmong [7]. Dietary supplements such as vitamin C (ascorbic acid) [13–15], cherry juice [16–20] or other supplements [21], represent attractive treatment options for those patients who resist taking prescription medications on a chronic basis [22].

Among the dietary supplements investigated to lower SU, vitamin C represents a promising option. Studies have demonstrated that vitamin C causes significant uricosuria by inhibiting reabsorption of renal urate in renal tubule in patients with gout [23–26]. Its mechanism is likely through the inhibition of transporters on proximal renal tubule such as URAT1 [27] and/or SLC5A8 and SLC5A12 [28]. A meta-analysis of 13 randomized controlled trials with 556 healthy adults found that the median dose of vitamin C 500mg daily for a median of 30 days was effective in lowering SU (0.35 mg/dL, 95%CI: 0.66–0.03, $p$-value = 0.032) compared to placebo [14]. A similar result was reported in a separate meta-analysis with a pooled sample size of 1,013 participants [29]. Furthermore, an epidemiology study with 46,994 male participants found individuals consuming 1000 to 1499 mg/day had a 45% (95%CI: 20%-62%)

decreased risk of gout in comparison to those consuming less than 250 mg/day [30]. However, a study in patients with gout receiving vitamin C 500mg once daily for 8 weeks failed to find a significant reduction in SU compared to the known effective prescription medicine allopurinol (0.23 mg/dL versus 1.9 mg/dL, $p$-value <0.001) despite a significant increase in plasma ascorbic acid level [13]. Collectively, these observations of relative effectiveness of vitamin C to lower SU, raise the prospect of identifying characteristics of individuals who are optimal responders to this intervention.

Vitamin C appears to exhibit linear kinetics at a lower dose and becomes non-linear at a higher dose to the saturable active transport mechanism in the absorption, distribution, and excretion. The maximum steady-state plasma concentration of ascorbic acid reaches about 70–80 uM [31]. The selection of vitamin C 500mg twice daily was to maintain steady-state concentration of ascorbic acid below the maximum concentration. Based on the pharmacokinetics of vitamin C (clearance of 6.02 L/hr and assuming the oral bioavailability of 1) [32], the estimated steady-state concentration of vitamin C 500mg is about 40 uM. Therefore, we choose a dose that was twice of the previous tested dose as the interventional dose but also one that is consistent with other previously conducted studies.

Given recent advancements in bioinformatic tools that enable the identification of taxonomic and functional patterns of gut microbiota and genetic variants, we may be able to identify appropriate medications and doses for various diseases, including metabolic diseases. Two examples include using individual's personal microbiome to predict glucose response to specific foods [33] or using genetic variants such as single nucleotide polymorphisms (SNPs) to predict SU-lowering response of allopurinol [34–37]. For example, Chinese participants with gout had higher abundances of intestinal microbiota in *Prevotella*, *Fusobacterium*, and *Bacteroides* yet lower in *Enterobacteriaceae* and *butyrate*-producing species compared to healthy adults [38, 39]. The functional analysis also showed participants with gout were enriched in genes for fructose, mannose metabolism and lipid A biosynthesis but low in genes for urate degradation and short chain fatty acid production [39]. In patients with gout, urate-lowering therapies such as allopurinol treatment for more than 30 days were found to reduce *Bacteroidaceae* and enhance *Ruminococcacea-Prevotellaceae* compared to those without allopurinol treatment [40]. In addition, vitamin C 1000mg daily for 2 weeks have shown a positive bacterial shift in healthy volunteers [41]. Based on these studies, we selected a 500mg twice daily dose of vitamin C for our study having several sources of evidence for efficacy for individuals with hyperuricemia/gout, while displaying a favorable tolerance or safety profile for those without hyperuricemia or gout.

We conducted the **M**icrobiome **AN**d **G**out, **O**besity, Vitamin **C** (HMANGO-C) study as a prospective open-labelled clinical trial whose primary objective was to evaluate the SU-lowering effect of vitamin C 500mg twice daily for 8 weeks in a high-risk Hmong adults. The secondary objectives were to assess if 1) vitamin C administration impacts the taxonomic and functional patterns of gut microbiota; 2) taxonomic and functional patterns of gut microbiota impact vitamin C's urate-lowering effects; 3) genetic variations in select genes impact vitamin C's urate-lowering effects; 4) differential microbial biomarkers exist for patients with or without gout and 5) there is an association between obesity, gut microbiota and gout. We utilized a community-based participatory research (CBPR) approach given the study's focus on an underserved unique population at risk.

## Materials and methods

### Design

HMANGO-C is a phase II, open-label, cross- sectional and interventional study, conducted with Hmong community partners during restrictions for face-to-face research in place for

safety from SARS-CoV-2 (COVID-19). The University of Minnesota Institutional Review Board approved the study (UMN IRB, STUDY00010406, approved on January 26, 2021) with a category of no greater than minimal risk and waived the requirement to document consent. Nonetheless, we obtained participant's consent prior to their beginning the study so we could ensure that people understood the study. The study received the exemption from the investigational new drug regulations of U.S. Food and Drug Administration. The study protocol was registered and approved on ClinicalTrials.gov (NCT04938024). Any protocol modifications including changes to eligibility criteria, outcomes, analyses will be submitted to UMN IRB. Upon approval, information will be updated on ClinicalTrials.gov.

## Community-based participatory research

Community-based participatory research (CBPR) is an approach to do research that engages community members and academic researchers in egalitarian partnerships in order to work collaboratively to identify and implement a research design that tries to solve a health challenge of community importance—in this case, gout in the Hmong community. Engaging community members as equal partners in the design and implementation of an intervention has been shown to demonstrate positive health outcomes in minority communities and other disadvantaged populations [42]. Our research design, recruitment and enrollment approaches were guided by Hmong community members to optimize effectively engage Hmong individuals residing in, or close to Minneapolis/St. Paul, metropolitan area.

## Study population

We plan to enroll 180 Hmong adults, aiming for 2:1 of participants with gout/hyperuricemia and participants without gout/HU. Target recruitment of 180 participants was chosen from predictions necessary to reach statistical power based on the predicted reduction of SU from vitamin C (see *Statistical Analysis* section for detail).

*Inclusion Criteria*

1. Individuals who self-identify as Hmong, attesting that both of their parents are Hmong

2. Eighteen years of age or older

*Exclusion Criteria*

1. Allergy or sensitivity to vitamin C

2. Diagnosis/history of:

   - Gastrointestinal surgery including colectomy, ileectomy, and gastrectomy

   - Inflammatory bowel disease

   - Auto-immune disease

   - Type I diabetes mellitus

   - Severe kidney disease (i.e., on dialysis)

   - End-stage liver disease (i.e., cirrhosis)

   - Pregnant women

   - Breastfeeding women

   - Glucose-6-phosphate dehydrogenase deficiency

3. Currently ingesting or had taken in the past 7 days

- Antibiotics

- Probiotics supplement

- Ketogenic diet

## Recruitment

Participant recruitment began in March 2021 and is ongoing. Our recruitment strategies included face-to-face and remote community outreach initiatives in order to promote safety during COVID-19 restrictions. We explained the study's importance to the community and encouraged people to enroll via presentations at Hmong organizations, Hmong television and radio stations, and social media platforms (e.g., HMANGO-C study website (URL: https://www.hmangoc.org/), Facebook page and YouTube videos). (See Supplemental material for recruitment flyers.) Interested people were instructed to contact our bilingual study staff.

## Enrollment

Bilingual Hmong community researchers explained the study purpose and specific requirements with potential participants by video call, phone call, and/or in-person meeting in order to obtain complete understanding and consent.

Fig 1 shows the schedule of enrollment, interventions, and assessments for the study. After participants were enrolled remotely by community researchers, collection kits were mailed to

| | STUDY PERIOD | | | | | | |
|---|---|---|---|---|---|---|---|
| | Enrolment (day) | Allocation | Post-allocation (weeks) | | | | Close-out (weeks) |
| TIMEPOINT** | -30 to -1 | 0 | 1 | 4 | 7 | 8 | 8 |
| **ENROLMENT:** | | | | | | | |
| *Eligibility screen* | X | | | | | | |
| *Informed consent* | X | | | | | | |
| *Privacy statement (HIPAA)* | X | | | | | | |
| **Allocation** | | X | | | | | |
| **INTERVENTIONS:** | | | | | | | |
| *Vitamin C 500 mg twice daily* | | | ◆———————————◆ | | | | |
| **ASSESSMENTS:** | | | | | | | |
| *Demographics, Medical history[1]* | | X | | | | | |
| *Anthropometrics[2]* | | X | | | | | X |
| *Vital signs* | | X | | | | | X |
| *Sample collection[3]* | | X | | | | | X |
| *Medication adherence* | | | X | X | X | | X |
| *Adverse effects* | | | X | X | X | | X |

**Fig 1. Schedule of enrollment, interventions, and assessments for the Hmong Microbiome ANd Gout, Obesity, Vitamin C (HMANGO-C) study.** (Adopted from SPIRIT 2013 Figure.), [1]Include pregnancy status, COVID-19 vaccine status, and dietary history, [2]Include height, weight, and waist circumference, [3]Two stool samples in two consecutive days, a clean-catch urine sample, a saliva sample, and a blood sample.

participants' homes. Participants self-collected stool, urine, and saliva samples, as explained by verbal instructions, written instructions, and videotapes. Participants arrived at a well-known community site, where we collected blood and anthropometry measurements, collected their samples and handed-out their vitamin C bottle.

## Data collection

Participants underwent the following assessments/ data collection:

**Health survey.** All study participants answered personal histories including lifestyle activities (diet, tobacco use, and exercise), family history, personal medical history, and medication history (including prescription, over-the-counter medications, probiotics, nutraceuticals and herbal therapies).

**Medication history.** Participants recorded their current medications and answered general questions about their adherence to these medicines. If deemed necessary, the researchers asked participants to either bring the medications they were taking to an in-person study visit or grant permission to investigators to acquire a current medication list from their medical clinic.

**Food survey.** All participants performed a comprehensive 1-month food recall history using a dietary history questionnaire (DHQ) III at the beginning or conclusion of the study. Participants with gout were prompted to identify their gout flare trigger foods, first by an open-ended question and then by reviewing a list of common trigger foods. Additionally, participants noted the amount of the trigger food and rank ordered the likelihood of these foods to trigger a gout episode.

**Gout assessment.** Individuals with gout completed a gout symptom online assessment tool using an established swollen joint count tool: https://www.carearthritis.com/tools/tools_html.anonlaunch?toolid=8&refid=/physicians.php%23tab2.

In addition, participants with gout also completed a modified Gout Assessment Questionnaire version 2.0 (GAQ 2.0) [43]. The GAQ 2.0 is a patient-reported outcome measurement tool with 24 items that assesses overall gout concerns, gout medication side effects, unmet treatment goals, wellbeing during gout attacks, and concerns during gout attacks. Since validation of this tool was conducted in a population with low ethnic diversity, we conducted a "cross-cultural adaptation of scales" process [44, 45] to evaluate the relevance of questionnaire's items for the Hmong community. The process yielded some changes in wording in order to retain the intent of the question, given cultural and linguistic context. For example, we expanded the inquiring into people's well-being during gout attacks, from focusing on work and recreational activities already assessed by the survey to asking about social obligations since the Hmong community values social obligations as equally important as other aspects of life. Also, we rephrased "gout symptoms" to "gout attack" because Hmong patients recognize the phrase "gout attack" but not necessarily "gout symptoms". The resulting product was translated into Hmong. The English version of the modified and translated Hmong GAQ 2.0 tools are available as supplementary material.

**Anthropometric and other measurements.** We measured participants' height, weight, waist, systolic and diastolic blood pressure, and heart rate at visits 1 and 2. Systolic and diastolic blood pressure and heart rate were measured in two arms each then averaged.

**Blood and urine biochemistry tests.** We collected blood and urine using plasma collection tubes with sodium heparin (BD Vacutainer®, BD, Franklin Lakes, NJ, USA) and 120 mL sterile urine cups, (Globe Scientific, Mahwah, NJ, USA). We will use the blood and urine samples to measure urate and creatinine as critical markers, and SARS-CoV-2 antibody.

**Genomic analysis.** Participants collected saliva samples using ORAgene® OGR-600 (DNA Genotek Inc., Ottawa, ON, Canada) based on the procedures described in the

*Biospecimen Collection* section below. From saliva samples, we will purify and extract genomic DNA using QIAamp DNA Kit (Qiagen Inc., Germantown, MD, USA) at the University of Minnesota Genomics Center (UMGC). We plan on analyzing genomic DNA for select genetic variants within genes such as *ABCG2* and *SLC22A12*, which have known associations with HU, gout, and the responses of ULT. Other genes and variants may be examined should new associations with these outcomes be discovered and identified from emerging literature [34, 46–48].

**Microbiome analysis.** All participants collected stool samples using a 1 mL stool tube with 95% ethanol and glass beads with Feces Catcher (Fecesvanger, Zeijen, Netherlands) and 20 selected participants (10 with gout and/or HU and 10 without gout) provided an additional fresh frozen stool sample. Microbiota DNA will be extracted from stool samples and be sequenced using whole-genome shotgun technique following the UMGC protocol [49]. Whole-genome shotgun sequencing data will be trimmed and processed for quality using SHI7 [50] and strain-level taxa will be annotated by comparison to a public strain database. Microbiome features (alpha diversity, beta diversity, taxon relative abundance, gene functional category) will be analyzed, as described in a previous study [51].

## Biospecimen collection

**Stool, saliva and urine collection.** Participants self-collected their stool, saliva and urine using respective collection kits mailed to their home address. Detailed collection instructions were located on a study website (https://www.hmangoc.org/sample-collection-instructions) in English and Hmong. Within two days prior to Visit 1, participants collected two stool samples in 95% ethanol preparation in two consecutive days, one saliva sample and one urine sample. Selected participants (10 from non-gout/HU group and 10 from gout/HU group) also collected fresh stool. Within two days prior to Visit 2, all participants collected two stool samples in two consecutive days and one urine sample. All self-collected samples were stored at room temperature.

**First in-person visit (Visit 1).** We held in-person visits to locations (e.g., healthcare clinics) known to the community; at the beginning of the pandemic, we met inside a UMN community research van; as restrictions loosened, we met inside buildings. Our use of a mobile clinical van created a "safer" environment limiting in-person interactions by avoiding the participants from having to enter a physical clinic setting during the COVID-19 pandemic.

At the events, phlebotomists performed venipunctures, researchers gathered participants' self-collected samples (urine, stool, and saliva), obtained anthropometrics (height, weight, and waist- circumference) and measured vital signs (blood pressure and heart rate). Once collected, blood samples were stored at 4°C (for less than 24 hours) until processing and testing at laboratory. Upon receipt of participants completed samples, participants obtained vitamin C 500mg capsules sufficient for 8 weeks of treatment and a $75 gift card.

**Second in-person visit (Visit 2).** Eight weeks later, we held the second in-person sample collection visit in the same manner as Visit 1, gathered participants' self-collected samples, measured anthropometrics and vital signs, and gave a $75 gift card.

## Treatment

Pure Encapsulations (Sudbury, MA, USA), a GMP certified manufacturer, labeled and delivered Vitamin C capsules (lot # 50241079). We instructed participants to take vitamin C 500mg twice daily by mouth for 8 weeks, along with their usual prescribed medications (including ULT), supplements and nutraceuticals during the study period. To facilitate adherence, we asked participants to record their vitamin C taking history in a study medication diary sheet

(See Supplemental Material), and we sent follow-up reminder phone/text/emails at weeks 1, 4 and 7, at these timepoints, we also documented study medication adverse effects as well as document any gout attacks which may have occurred during vitamin C treatment period. To assess study medication adherence, we used the medication diary sheet recorded by participants at Visit 2. The assessment of blood levels of ascorbic acid after treatment to verify the medication adherence is not planned because ascorbic acid is rapidly degraded [52] and current study design did not allow a reliable measurement of ascorbic acid.

## Analysis

**Assignment of membership to gout/hyperuricemia group.** Individuals with gout were defined as those with self-reported gout history, with at least 1 episode of acute gout flare, or self-reported history of allopurinol or other ULT ingestion for gout, regardless of baseline SU. Individuals with HU were those with a baseline $SU \geq 6.8$ mg/dL.

**Sample size.** For an effect size of 0.5 mg/dL SU reduction from baseline by vitamin C [14], using a one-sided *t*-test, 80% power, type 1 error of 0.01, and 15% drop-out rate, 120 in adults with gout/HU (standard deviation, SD of 1.5 mg/dL) [17] and 60 adults without gout/HU (SD of 1 mg/dL) were required [12].

**Analysis plan.** Absolute SU changes from baseline to week 8 will be calculated using multiple linear regression adjusting for patients' characteristics including age, gender, BMI, renal function, and concomitant medications. The impact of genetic variants and microbiome features will be tested utilizing stepwise regression.

Overall differences in microbiome composition from baseline to week 8 will be compared using three principal axes of variation in the microbiome. A linear regression will be analyzed with a bacterial taxonomy as a dependent variable and host factors as independent variables. The significance of association of a host factor will be determined by the test of the null hypothesis that the coefficient is equal to zero. For microbiome features, we will correct computed p-values for the total number of comparisons between dependent and independent variables using false discovery rate correction.

## Data collection and management

Baseline characteristics, including participants' demographics, anthropometrics, medical/medication history, and vital signs, were collected remotely via REDCap survey or in-person at respective clinical visits. Blood and urine samples were sent to Advanced Research and Diagnostic Laboratory (ARDL), University of Minnesota for analysis. Saliva and stool samples were sent to UMGC for analysis.

The study consent process provided participants the option to have samples saved and analyzed for future research questions. For those participants who agreed, their DNA and any stool, urine, or blood samples remaining after completed primary analysis will be stored in secured UMN -80 freezers for future studies.

Participants' identifiers were available only to members of the research team throughout the data collection period to contact participants for safety and study medication protocol adherence. At the end of the data collection period, participants' key identifiers and their unique study ID will be maintained on a full disk encrypted secure server behind the UMN firewall, with access only by the Principal Investigator (PI) and co-PIs. We will access participants' identifiers to invite them to a future dissemination event.

The digital data was entered into a REDCap database, which uses a MySQL database via a secure web interface with data checks used during data entry to ensure data quality. Participant accrual data was entered into OnCore as required by the University of Minnesota's Academic

Health Center. OnCore is a suite of clinical and translational research modules consisting of software for research, patient registry, and biospecimen management.

## Dissemination

Upon completion of the study, study participants will receive individual results including vital signs, anthropometry, renal function, serum urate level, COVID19 antibodies, genetics, and microbiome results. Aggregated genetics and microbiome results will also be shared with participants and the Hmong community at several venues and through several media. These may include locations, such as recruitment sites, community conferences, community events, community organizations and media including Hmong radio and television, and the Hmong Gout Coalition website. In addition, we will share results with local physicians and pharmacists who have partnered with recruitment, and we will disseminate our results with broader audiences by presenting at local, national, and international conferences, and by publishing peer-reviewed articles in scientific journals.

Individual participant results will also be shared at the conclusion of the study. We will mail relevant individual results to the participants. Since the genetic results will not affect routine clinical care, credentialed genetic professionals will not be used to explain study results in-person. However, we will stress that analysis of biological samples is for research purposes only and are not necessary for participants' clinical care.

## Study adjustments due to the impact of COVID-19

The pandemic of severe acute respiratory syndrome coronavirus 2 (SARS-CoV-2), commonly referred to as COVID-19, required adjusting the initial planned research approaches and practices. Shortly after the onset of the pandemic (around January 2020), studies involving in-person visits were required to adopt protocol modifications to minimize the physical contact with study participants to avoid the spread of SARS-CoV-2. As a result, our study protocol, which was designed to consider the unique challenges of working in an underserved population, required further adaptations to comply with COVID-19 guidance for research. We planned to recruit and enroll participants remotely by telephone and video, complete questionnaires on the internet, accept patient-reported anthropometry, and have participants self-collect blood, urine, spit, and stool in the safety of their own homes.

With the nationwide implementation of COVID-19 vaccination plan (Spring 2021), it was anticipated that the COVID-19 restrictions regarding scientific research could be relaxed, such that in-person group screening, consenting and sample collecting events could occur. Throughout most of our data collection, we operated under an intermediate phase whereby we limited in-person visits were permitted to enable blood draws, anthropometry, and vital sign collection, while the majority of the recruitment processes, data collection of questionnaires, and biological sample collection occurred remotely.

Due to COVID-19 restrictions, participation engagement occurred remotely. Study participants' consent was obtained via an individual Zoom meeting or a direct conversation via a phone call, and demographic assessments were conducted remotely with virtual assistance provided by the study researchers. Stool, saliva, and urine samples were successfully collected at home. Early in the process (April 2021 through May 2021), at home blood collection using capillary blood collection tubes (200 uL and 500 uL) was attempted. However, obtaining enough blood for analysis from our first few participants failed, and had to be reconsidered. As vaccines became more widely available (Summer 2021), blood collection was changed to in-person visit as the restrictions for person interactions with participants were lifted. Biometric measurements were also conducted in person at a pre-determined convenient location by use

of a study van. This was pursued to enhance enrollment by providing a convenient location for members of this community to engage.

## Conclusion

This study protocol documents a number of unique and diverse approaches to conducting a clinical trial. First, we designed the study based on the community-based participatory research principles that focus on engaging, identifying, and proposing a plan to address an issue of concern that is important to members of the community and utilizing an academic-community partnership to design and conduct a study aimed at solving a community's health challenge. Second, we explored the effects of vitamin C as an intervention, analyzing its effect on the microbiome and omic-based markers which may predict its response. Third, we modified study procedures and approaches to meet the COVID-19 restrictions concerning limited in-person interactions at various times during the pandemic. We expect these results will provide insight as to the clinical utility of using microbiome and genetic information for patients with hyperuricemia and gout who could benefit from vitamin C supplementation. These results could provide guidance to clinicians who want to offer an accessible, acceptable, and low-cost treatment option for high-risk study population of Hmong. Conducting clinical research involving special populations—even during a pandemic—is necessary to advance knowledge about therapies for important clinical community problems. Several aspects of this study can assist academic-community research partnerships who conduct research on problems important to underserved communities.

## Supporting information

**S1 Checklist. SPIRIT checklist.**
(PDF)

**S1 File. HMANGO-C study protocol.**
(PDF)

**S2 File. HMANGO-C consent form English.**
(PDF)

**S3 File. HMANGO-C HIPAA form English.**
(PDF)

**S4 File. HMANGO-C consent form Hmong.**
(PDF)

**S5 File. HMANGO-C HIPAA form Hmong.**
(PDF)

**S6 File. HMANGO-C study flyer English.**
(PDF)

**S7 File. HMANGO-C study flyer Hmong.**
(PDF)

**S8 File. Gout assessment questionnaire 2.0 Hmong translation.**
(PDF)

**S9 File. Example of study medication diary sheet.**
(PDF)

## Acknowledgments

The authors wish to thank the support from all the study participants, members of Hmong Gout Coalition, the community advisory board: Jay Desai, Song Xiong, Lissee Thao, Maiyia Kasouaher, and Mai See Vang Moua, and members of SoLaHmo: Michelle Lo, Pa Jee Vang, and Colton Vue who provided important input and the recruitment of the participants. We also thank Charles Vang, a member of the Hmong 18 Council, who provided critical feedback on the study design and helped the promotion of the study.

## Author Contributions

**Conceptualization:** Ya-Feng Wen, Kathleen A. Culhane-Pera, Muaj Lo, Dan Knights, Robert J. Straka.

**Data curation:** Ya-Feng Wen, Yeng Moua, Bai Vue, Toua Yang, Boguang Sun.

**Funding acquisition:** Dan Knights.

**Investigation:** Ya-Feng Wen, Kathleen A. Culhane-Pera, Boguang Sun, Dan Knights, Robert J. Straka.

**Methodology:** Ya-Feng Wen, Kathleen A. Culhane-Pera, Shannon L. Pergament, Yeng Moua, Bai Vue, Toua Yang, Muaj Lo, Boguang Sun, Dan Knights, Robert J. Straka.

**Project administration:** Ya-Feng Wen, Kathleen A. Culhane-Pera, Shannon L. Pergament, Boguang Sun, Dan Knights, Robert J. Straka.

**Supervision:** Kathleen A. Culhane-Pera, Robert J. Straka.

**Writing – original draft:** Ya-Feng Wen, Kathleen A. Culhane-Pera, Shannon L. Pergament, Boguang Sun, Dan Knights, Robert J. Straka.

**Writing – review & editing:** Ya-Feng Wen, Kathleen A. Culhane-Pera, Shannon L. Pergament, Yeng Moua, Bai Vue, Toua Yang, Muaj Lo, Boguang Sun, Dan Knights, Robert J. Straka.

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
