## [Decision Letter · Decision Letter 0]

10 Aug 2022

PONE-D-22-04805

Hmong Microbiome ANd Gout, Obesity, Vitamin C (HMANGO-C): A phase II clinical study protocol

PLOS ONE

Dear Dr. Straka,

Thank you for submitting your manuscript to PLOS ONE. After careful consideration, we feel that it has merit but does not fully meet PLOS ONE’s publication criteria as it currently stands. Therefore, we invite you to submit a revised version of the manuscript that addresses the points raised during the review process.

We look forward to receiving your revised manuscript.

Kind regards,

Cristina Vassalle

Academic Editor

PLOS ONE

https://journals.plos.org/plosone/s/file?id=ba62/PLOSOne_formatting_sample_title_authors_affiliations.pdf".

“Minnesota Partnership for Biotechnology and Medical Genomics grant #18.08, Targeting the gut microbiome to prevent the increasing incidence of obesity in immigrant populations. This funding source had no role in the design of this study and will not have any role during its execution, analyses, interpretation of the data or decision to submit results.”

 “D.K received funding from  Minnesota Partnership for Biotechnology and Medical Genomics grant #18.08, Targeting the gut microbiome to prevent the increasing incidence of obesity in immigrant populations. This funding source had no role in the design of this study and will not have any role during its execution, analyses, interpretation of the data or decision to submit results.”

“D.K. serves as CEO and holds equity in CoreBiome, a company involved in the commercialization of microbiome analysis. The University of Minnesota also has financial interests in CoreBiome under the terms of a license agreement with CoreBiome. These interests have been reviewed and managed by the University of Minnesota in accordance with its Conflict-of-Interest policies.”

5. We note that the original protocol file you uploaded contains a confidentiality notice indicating that the protocol may not be shared publicly or be published. Please note, however, that the PLOS Editorial Policy requires that the original protocol be published alongside your manuscript in the event of acceptance. Please note that should your paper be accepted, all content including the protocol will be published under the Creative Commons Attribution (CC BY) 4.0 license, which means that it will be freely available online, and any third party is permitted to access, download, copy, distribute, and use these materials in any way, even commercially, with proper attribution.

Therefore, we ask that you please seek permission from the study sponsor or body imposing the restriction on sharing this document to publish this protocol under CC BY 4.0 if your work is accepted. We kindly ask that you upload a formal statement signed by an institutional representative clarifying whether you will be able to comply with this policy. Additionally, please upload a clean copy of the protocol with the confidentiality notice (and any copyrighted institutional logos or signatures) removed.

Reviewers' comments:

Reviewer's Responses to Questions

**Comments to the Author**

1. Does the manuscript provide a valid rationale for the proposed study, with clearly identified and justified research questions?

Reviewer #1: Yes

Reviewer #2: Yes

2. Is the protocol technically sound and planned in a manner that will lead to a meaningful outcome and allow testing the stated hypotheses?

Reviewer #1: Yes

Reviewer #2: Yes

3. Is the methodology feasible and described in sufficient detail to allow the work to be replicable?

Reviewer #1: Yes

Reviewer #2: Yes

4. Have the authors described where all data underlying the findings will be made available when the study is complete?

Reviewer #1: Yes

Reviewer #2: Yes

5. Is the manuscript presented in an intelligible fashion and written in standard English?

Reviewer #1: Yes

Reviewer #2: Yes

6. Review Comments to the Author

You may also provide optional suggestions and comments to authors that they might find helpful in planning their study.

Reviewer #1: This is an interesting and well written paper; the protocol is carefully described.

Some comments.

In the introduction the possible role of Vit C on gut microbiota should be shortly added to better define point 1 of the secondary objective, e.g. in a healthy population high doses of Vit C have shown a positive shift of bacteria populations in the gut (Otten 2021).

Similarly, in the introduction when dealing with metabolic diseases, published data on the presence of dysbiosis in gout patients and the role of specific urate-reducing drugs in modifying gut microbiota should be provided.

Are pts under diuretic treatment allowed in the study?

"To assess study medication adherence, we used the medication diary sheet recorded by participants at Visit 2". It should be discussed that the assessment of blood levels of ascorbic acid after treatment is not planned.

Reviewer #2: This study protocol is well structured and well elaborated., although there are some critical points that the authors should remedy.

The authors should better explain the rationale for the administration of vitamin C to subjects without gout/hyperuricaemia.

How were the doses of vitamin C to be administered chosen?

Among the primary objectives, the authors should also introduce placebo especially in the patient group.

In my opinion, when analysing the data, the authors should also take gender into account.

Fig. 1 is actually a table

As far as blood and urine tests are concerned, the authors want to study the urate-creatinine trend.

It would also be good to analyse markers of oxidative stress, such as ROS, or indices of biological oxidative damage such as malondialdehyde and isoprostanes.

Vitamin C at high plasma concentrations may act as a pro-oxidant (1) Have the authors ever tested this effect or is there any data for the treatment they want to perform? Any treatment with vitamin C in patients genetically predisposed to high uric acid levels should be performed for life. According to the authors, would this not lead to an increased oxidative risk in these subjects? (2)

1) Julia Kaźmierczak-Barańska , Karolina Boguszewska, Angelika Adamus-Grabicka

Bolesław T. Karwowski. Two Faces of Vitamin C—Antioxidative and

Pro-Oxidative Agent.Nutrients 2020, 12, 1501; doi:10.3390/nu12051501

2) Ian D. Podmore, Helen R. Griffiths, Karl E. Herbert, Nalini Mistry, Pratibha Mistry & Joseph Lunec. Vitamin C exhibits pro-oxidant properties. NATURE | VOL 392 | 9 APRIL 1998

7. PLOS authors have the option to publish the peer review history of their article (what does this mean?). If published, this will include your full peer review and any attached files.

Reviewer #1: No

Reviewer #2: No

---

## [Author Response · Author response to Decision Letter 0]

16 Sep 2022

Response to reviewers has been uploaded as a separate file

---

## [Decision Letter · Decision Letter 1]

20 Oct 2022

PONE-D-22-04805R1Hmong Microbiome ANd Gout, Obesity, Vitamin C (HMANGO-C): A phase II clinical study protocolPLOS ONE

Dear Dr. Straka,

Thank you for submitting your manuscript to PLOS ONE. After careful consideration, we feel that it has merit but does not fully meet PLOS ONE’s publication criteria as it currently stands. Therefore, we invite you to submit a revised version of the manuscript that addresses the points raised during the review process.

We look forward to receiving your revised manuscript.

Kind regards,

Cristina Vassalle

Academic Editor

PLOS ONE

Additional Editor Comments:

Please, answer to reviewer 2

Most of the comments made were not taken into account. The authors state that they have difficulties in introducing other markers or the placebo due to their limited budget.

In my opinion, there are aspects not to be underestimated in this study, such as the production of oxygen free radicals in patients subjected to high doses of vitamin C for long periods.

Solving the problem of Fig/Tab. 1

Reviewers' comments:

Reviewer's Responses to Questions

**Comments to the Author**

1. Does the manuscript provide a valid rationale for the proposed study, with clearly identified and justified research questions?

Reviewer #1: Yes

Reviewer #2: Partly

2. Is the protocol technically sound and planned in a manner that will lead to a meaningful outcome and allow testing the stated hypotheses?

Reviewer #1: Yes

Reviewer #2: Yes

3. Is the methodology feasible and described in sufficient detail to allow the work to be replicable?

Reviewer #1: Yes

Reviewer #2: Yes

4. Have the authors described where all data underlying the findings will be made available when the study is complete?

Reviewer #1: Yes

Reviewer #2: Yes

5. Is the manuscript presented in an intelligible fashion and written in standard English?

Reviewer #1: Yes

Reviewer #2: Yes

6. Review Comments to the Author

You may also provide optional suggestions and comments to authors that they might find helpful in planning their study.

Reviewer #1: I thank the AA for their responses. For me, the paper is ready for publication in its present form since all my questions have been answered in a satisfactory manner

Reviewer #2: Most of the comments made were not taken into account. The authors state that they have difficulties in introducing other markers or the placebo due to their limited budget.

In my opinion, there are aspects not to be underestimated in this study, such as the production of oxygen free radicals in patients subjected to high doses of vitamin C for long periods.

Solving the problem of Fig/Tab. 1

7. PLOS authors have the option to publish the peer review history of their article (what does this mean?). If published, this will include your full peer review and any attached files.

Reviewer #1: No

Reviewer #2: No

---

## [Author Response · Author response to Decision Letter 1]

23 Nov 2022

Thank you for your feedback concerning our comprehensive response to the initial review. We are pleased that each of the reviewers took time to evaluate our responses and for the most part appreciated our responses as honest attempts to address their valued feedback. We are also happy to respond to additional comments made by the Reviewer 2. 

Specific to Reviewer 2, we respectfully disagree with reviewer’s feedback on “most of the comments made were not taken into account”. Instead, we believe that most of the comments from Reviewer 2 (see Comments 1-6 in the section of Reviewer 2 Comment from First Revision) had been adequately addressed and incorporated into manuscript since Reviewer 2 did not have further suggestions on these comments. 

Regarding reviewer 2’s suggestion on testing biomarkers of oxidative stress or DNA damage such as 8-oxoguanine and 8-oxoadenine (Kaźmierczak-Barańska et al. and Podmore et al.) due to increased oxidative risk from high dose vitamin C (500 mg twice daily), we have the following rebuttal. As previously described in the paper, the aim of our study is to evaluate the efficacy (by measuring serum urate change) and safety (by collecting patient reported adverse events) of vitamin C over the course of 8 weeks (not chronically) on the microbiome within and between individuals with and without gout and/or hyperuricemia. 

The two key points here are (1) that we are looking for a signal from 8 weeks of vitamin C exposure (not chronic therapy as the reviewer suggested) and that (2) the dose chosen (1g/day) was by no means excessive, based on a variety of studies conducted in a variety of subjects using, in some cases much higher doses over similar, and in some cases, longer timeframes than our 8 weeks. Clinical studies have demonstrated the safety profile of a wide dose ranges of vitamin C in various populations (Yanase et al. Pediatr Crit Care Med. 2021;22(6):561-571; Andrés et al. Cochrane Database Syst Rev. 2021;11(11):CD010156.). Therefore, we are confident that the dose tested in the study is safe for the study participants in our setting. In addition, as for the suggestion to incorporate testing of biomarkers of oxidative stress or DNA damage in our study, we feel as stated before that the need for this is outside the scope of our study’s objectives. Including additional biomarkers would require a protocol change, revisions of informed consent, and the need to re-contact study participants which is not feasible given the study timeline of this study and resources. To reiterate our previous response, if we identify a favorable effect of Vitamin C on serum urate in our study population for our (8 week) short course of treatment, and a link between the microbiome and urate as well, we would naturally pursue more carefully designed dose ranging studies which would specifically test for the “optimal” dosages for the positive effects of vitamin C and any possible untoward effects (including pro-oxidative effects for example) especially if we pursued longer term (>8 week) use. It would also evaluate whether indeed, chronic dosing of vitamin C would be necessary or episodic (non-chronic use) of vitamin C would be more favorable. Since these are all possibilities it seems as if we should await a signal before planning the next series of studies. 

We do note that Reviewer 2’s comments represent a statement of “In my opinion” and generally agree with the potential value in further study, but simply do not feel that Reviewer 2’s opinion justifies a change in our protocol at this stage, in order to investigate the possibility of a signal for the objective we are seeking. I will note that this is simply a pilot study seeking preliminary data in support of future more extensively designed investigations.

Reviewer 2 Comment from Second Revision

Most of the comments made were not taken into account. The authors state that they have difficulties in introducing other markers or the placebo due to their limited budget.

In my opinion, there are aspects not to be underestimated in this study, such as the production of oxygen free radicals in patients subjected to high doses of vitamin C for long periods.

As a result of our careful consideration of “Reviewer 2’s Comments” we believe we have addressed them as above in a manner that does not require changes to the current version of the manuscript posted. This is why you will not see an updated version of the manuscript but simply our rebuttal to Reviewer 2’s comments as above.

---

## [Editor Report · Decision Letter 2]

15 Dec 2022

Hmong Microbiome ANd Gout, Obesity, Vitamin C (HMANGO-C): A phase II clinical study protocol

PONE-D-22-04805R2

Dear Dr. Straka,

We’re pleased to inform you that your manuscript has been judged scientifically suitable for publication and will be formally accepted for publication once it meets all outstanding technical requirements.

Kind regards,

Cristina Vassalle

Academic Editor

PLOS ONE
---

## [Editor Report · Acceptance letter]

20 Jan 2023

PONE-D-22-04805R2 

Hmong Microbiome ANd Gout, Obesity, Vitamin C (HMANGO-C): A phase II clinical study protocol 

Dear Dr. Straka:

I'm pleased to inform you that your manuscript has been deemed suitable for publication in PLOS ONE. Congratulations! Your manuscript is now with our production department. 

Kind regards, 

on behalf of

Dr. Cristina Vassalle 

Academic Editor

PLOS ONE